# Sense of Agency and Skills Learning in Virtual-Mediated Environment: A Systematic Review

**DOI:** 10.3390/brainsci14040350

**Published:** 2024-03-31

**Authors:** Valentina Cesari, Sveva D’Aversa, Andrea Piarulli, Franca Melfi, Angelo Gemignani, Danilo Menicucci

**Affiliations:** 1Department of Surgical, Medical and Molecular Pathology and Critical Care Medicine, University of Pisa, 56126 Pisa, Italy; valentina.cesari@phd.unipi.it (V.C.); sveva.daversa@gmail.com (S.D.); andrea.piarulli@unipi.it (A.P.); franca.melfi@unipi.it (F.M.); angelo.gemignani@unipi.it (A.G.); 2Clinical Psychology Branch, Azienda Ospedaliero-Universitaria Pisana, 56126 Pisa, Italy

**Keywords:** virtual reality, procedural learning, motor embodiment, distortions, multisensory processing

## Abstract

Agency is central to remote actions, and it may enhance skills learning due to a partial overlap between brain structures and networks, the promotion of confidence towards a telemanipulator, and the feeling of congruence of the motor choice to the motor plan. We systematically reviewed studies aiming to verify the role of agency in improving learning. Fifteen studies were selected from MEDLINE and Scopus^®^. When a mismatch is introduced between observed and performed actions, the decrease in agency and learning is proportional to the intensity of the mismatch, which is due to greater interference with the motor programming. Thanks to multisensory integration, agency and learning benefit both from sensory and performance feedback and from the timing of feedback based on control at the goal level or the perceptual–motor level. This work constitutes a bedrock for professional teleoperation settings (e.g., robotic surgery), with particular reference to the role of agency in performing complex tasks with remote control.

## 1. Introduction

The control of remote actions such as those performed in virtual-mediated environments is favored by the sense of embodiment, namely by the ability to feel the body as one’s own and more importantly by incorporating objects in the subjective representation of the body [1]. The sense of embodiment is in turn defined by three primary components: (i) self-location, which involves the subjective feeling of being located in a remote space; (ii) ownership, the faculty of perceiving non-bodily objects, e.g., tools, as integral parts of one’s own body; (iii) agency, the sense of being in control of observed actions while voluntarily controlling movements [2,3].

In teleoperation, the paramount importance of agency becomes evident, as performing active movement sustains the execution of tasks characterized by significant uncertainties such as robotic-assisted surgery [3,4], in which surgeons utilize artificial arms controlled via immersive visors [5]. Success in robotic-assisted surgery hinges on surgeons perceiving these robotic arms as their extensions, emphasizing the significance of studying embodiment and agency in this field [6].

For individuals interacting with telemanipulators, both explicit (subjective judgment of agency) and implicit (feeling of control) senses of agency are pivotal for fostering natural and intuitive interaction [7,8,9], as an enhanced sense of agency towards remote manipulators improves dexterity by reducing susceptibility to variations in tool characteristics such as size, motion, and degrees of freedom [3]. These processes that lead to optimal integration between humans and effectors are sustained by multisensory integration, which combines temporally aligned signals from various sensory sources and plays a pivotal role in maintaining agency and embodiment [10,11,12].

Neurophysiological studies in virtual reality revealed a set of brain regions implicated in agency perception during teleoperation. The activation of posterior parietal regions, including the right angular gyrus, augments sensorimotor control over avatars [10], while a network involving the right supramarginal gyrus, the left anterior inferior parietal lobule, the anterior insula, and the right temporoparietal junction detects agency mismatches [11]. Prefrontal cortex involvement is crucial for conscious agency experiences [11]. Farrer and colleagues observed increased cerebral blood flow in the right inferior parietal lobule during VR-induced loss of agency, as well as reduced activity in the right posterior insula and cerebellum [12]. EEG studies by Bu-Omer and colleagues and Kang and colleagues indicate that increased alpha and beta bands are associated with decreased agency perception, while heightened control correlates with decreased alpha relative power and reduced phase coherence within frontal regions [13,14].

As agency is the most involved component of embodiment in visuomotor dexterous performance, an important branch of learning that may particularly benefit from an enhanced sense of agency is procedural learning, including learning of skills and motor–perceptual abilities [15]. In everyday life, individuals can automatically (or with minimal attention) perform several sequences of movements (motor programs); however, when individuals must perform telemanipulated actions, the aforementioned motor programs could be subjected to variations. It has been found that motor learning programs performed in virtual environments are slower and less accurate as compared to those performed in non-virtual environments; this could be ascribed to the lack of experience with telemanipulated tools, uncertainty about the real position of the manipulated objects, a restricted field of view due to the headset, and a putative distortion of the virtual space that implies visuomotor remapping [16,17].

The mismatch between skills learning performed in teleoperation environments and physical reality could be explained by advocating for the role of the sense of agency. Despite its importance in teleoperation environments, a characterization of the role of agency in skills learning is still lacking. However, it could be assumed that a sense of agency might enhance skills learning based on the following considerations:They show a partial overlap between brain structures and networks. The medial frontal cortex (including supplementary motor area), posterior insula, occipital lobe, and cerebellum form the agency network [18]. Motor skills learning includes corticostriatal and corticocerebellar circuits, involving important brain regions like the striatum, cerebellum, primary motor cortex, and supplementary motor area during initial phases [19,20]. After learning acquisition, decreased corticocerebellar circuit activity suggests intentional learning, while sustained striatal activation indicates long-term retention [19,20].The degree of the experienced sense of agency towards the telemanipulator, which could be mediated by multisensory integration and is essential for task completion in virtual reality, might foster visuomotor coordination, thus speeding up the process of skills learning [3].The sense of agency could define how motor choice conforms to motor plan, thus helping individuals to regulate motor behavior during visuomotor skills learning in situations of uncertainty [21].

Based on these premises, we tested whether and how the sense of agency might affect the individuals’ skills learning in mediated environments by performing a systematic review of the related scientific literature. This work could provide an in-depth comprehension of the role of the sense of agency in skills learning and potentially lay the foundation for future research in teleoperation, including applications such as robotic surgery.

## 2. Materials and Methods

This systematic review was conducted by the Preferred Reporting Items for Systematic Reviews and Meta-Analyses (PRISMA) guidelines [22]. The protocol of this review has been submitted (ID 516572) to the international prospective register for systematic reviews database PROSPERO. To identify an effective search strategy, we adhered to the Population, Intervention, Comparison, Outcomes, and Study Design (PICOS) worksheet (Table 1) [22].

The research was conducted up to 14 June 2023.

The three main steps performed for conducting the study are illustrated in the PRISMA flow diagram (Figure 1), and detailed below:

1. An automatic search was conducted, based on specific queries applied to the MEDLINE and Scopus^®^ databases (search extended up to June 2023), and no restrictions on publication date were applied. Keywords referring to sense of agency (“agency” OR “sense of agency” OR “motor embodiment” OR “embodiment”) and procedural learning (“implicit learning” OR “perceptual learning” OR “skill acquisition” OR “skill learning” OR “procedural learning” OR “implicit knowledge” OR skills OR “motor learning” OR “motor control” OR “sequence learning” OR “motor sequence learning”) were identified to perform the search across the databases. Full search terms are provided in Appendix A. Duplicate publications were identified and eliminated using the Mendeley desktop reference manager (http://www.mendeley.com, accessed on 13 July 2023);

2. The retrieved articles underwent manual screening based on inclusion/exclusion criteria (Table 1) defined a priori, assessing the title and abstract only. The restrictions regarding the inclusion of studies exploring the sense of agency and skill acquisition in mediated environments were applied at this stage;

3. An in-depth assessment of each resulting paper was conducted based on a full-text examination.

Two independent researchers [VC and SD] conducted the literature search and study selection. Any disagreement was resolved through discussion with an arbiter [DM]. We employed two distinct methods for assessing the quality of the included studies based on their design:The following criteria outlined by Ding and colleagues [23] (Appendix A) were used: (1) proper cross-over design; (2) randomized treatment order; (3) consideration of carry-over effects; (4) unbiased data collection; (5) allocation concealment; (6) blinding procedures; (7) handling of incomplete outcome data; (8) avoidance of selective outcome reporting; and (9) addressing any other potential sources of bias. The risk of bias regarding each criterion was categorized as low (−), unclear (?), or high (+).The Critical Appraisal Skill Programme (CASP) checklist comprises several sections. The initial three questions (Section A) are screening questions used to assess the validity of the basic study design. If the study design passes this initial assessment, then it is evaluated for methodological soundness in Section B. The appraisal then continues with the remaining questions in Sections C and D. The study is assessed by selecting responses of “Yes”, “No”, or “Can’t tell” (Appendix A) [24]. As we also included non-randomized control trial studies, we omitted question numbers 2, 3, and 4, according to the indications given by the algorithm for classifying study design based on their effectiveness (indications found at https://www.sign.ac.uk/).

## 3. Results

### 3.1. Study Selection and Characteristics

The search process for each database is outlined in Appendix A. Articles retrieved from PubMed and Scopus databases were merged into a list comprising 2010 papers. This initial pool underwent a screening based on predetermined inclusion/exclusion criteria, resulting in 15 papers for qualitative synthesis. These papers collectively included data from 428 subjects (Figure 1).

#### 3.1.1. Sample Size

The included studies showed different sample size dimensions; the total number of recruited participants is 428, ranging from a minimum of 9 volunteers to a maximum of 40.

#### 3.1.2. Experimental Design

Among the collected studies, fourteen studies used a within-subject experimental design [25,26,27,28,29,30,31,32,33,34,35,36,37,38], whereas one study used a between-subjects experimental design [39].

#### 3.1.3. Outcomes and Measures of Agency

Thirteen studies reported behavioral outcomes [26,27,28,29,30,31,32,33,35,36,37,38,39], whereas two studies also included results on psychophysiological correlates [25,34].

Regarding the type of agency, five studies investigated implicit agency [25,27,28,29,31], whereas the other ten investigated a mixed sense of agency (implicit and explicit) [25,26,30,32,33,34,36,37,38,39].

#### 3.1.4. Type of Experimental Paradigms and Setups

The studies differed in the type of mediated environment: nine studies used immersive virtual reality settings [25,26,27,28,31,32,34,36,37], whereas six used non-immersive virtual reality (subjects were not placed in a three-dimensional scenario using headsets) [29,30,33,35,38,39].

To study the role of agency, two different paradigms were used as modulators: mismatches and experimental feedback.

Eleven studies implemented a paradigm based on mismatch, consisting of a spatial mismatch (virtual hand was shifted relative to the real one) [28,30,34,35], temporal mismatch (virtual hand moved with a delay ranging from 15 to 900 ms, with a mean value of 305 ms) [32,36,37,38], or physical mismatch (such a disturbing motor control by using the added noise of the physical weight of the end-effectors) [26,29,33].

Five paradigms were based on experimental feedback, consisting of sensory (haptic; visual) or performance feedback used to observe their impact on the sense of agency and skills learning: two studies introduced performance feedback (such as a positive explicit confirmation of the performance) [29,31], one study used visual feedback (by using virtual avatar providing real-time information) [25], and the other two introduced haptic feedback (such as a haptic return during billiard paradigms and haptic guidance with fixed trajectories during motor performance) [27,39].

#### 3.1.5. Quality Assessment of Included Studies

According to the risk of bias in within-subject studies measured using the items proposed by Ding and colleagues [23], the selected studies showed several biases regarding blinding procedures (high risk of bias was detected) [25,26,27,28,29,30,31,32,33,34,35,36,37,38], whereas information regarding carry-over effect, allocation concealment, and blinding procedures was missing, thus rendering the qualification of the risk of bias unclear (Appendix A). For what concerns the Critical Appraisal Skill Programme (CASP) of randomized controlled trials, the included studies showed a medium quality (Appendix A) [39]. A detailed assessment of the quality of each study is presented in Appendix A.

### 3.2. Synthesis of the Main Findings

We grouped the main findings of the included studies according to the experimental paradigm (mismatch versus experimental feedback). The main findings (statistical significance *p* < 0.05, for results below) are synthesized in the following sections.

#### 3.2.1. Studies Based on Motor Mismatch

Studies [26,28,29,30,32,33,34,35,36,37,38] introduced motor mismatch; participants perceived virtual movement as congruent or not with their effective action depending on spatial, temporal, or physical mismatch (Table 2, Figure 2A, Appendix A). Among the experimental manipulations, three main types of mismatch could be identified:Spatial mismatch, in which participants perceived a visuospatial change between the performed and virtually seen action;Temporal mismatch, in which participants were subjected to a temporal delay between the performed and virtual action;Physical mismatch, in which participants perceived an additional noise that made actions more difficult to perform.

##### Spatial Mismatch

Ratcliffe and Newport used a virtual copy of the real hand (with alterations in its appearance, movement, and location) to induce changes in visual feedback during the performance of tapping or pointing tasks [35]. The subjective evaluation of the performance and sense of agency were also measured. The sense of ownership was influenced by appearance, the location of the virtual hand, and synchrony between the real and virtual hands. The sense of agency was instead influenced only by synchrony, regardless of location and appearance.

Kannape and colleagues used a visuomotor conflict, altering the orientation and direction of movement in two experiments [28]. In Experiment 1, the walking trajectory of the virtual body was deviated during locomotion; in Experiment 2, the virtual body was represented from four points of view (upright/back, upright/front, inverted/back, and inverted/front). Experiment 1 showed a negative association between participants’ walking trajectory endpoints and the magnitude of the angular deviation applied to the virtual body’s walking trajectory. In Experiment 2, motor awareness, identifying the synchronization between the virtual body’s movement and the participant’s actual movement, did not vary significantly. However, motor performance, which represents the total angle compensated by the participant, exhibited higher scores in the upright/back condition as compared to the upright/front and inverted/back conditions.

Padrao and colleagues conducted two experiments, in which the virtual hand’s movements were congruent or incongruent with the real movements during the performance of a task requiring visuomotor and attentional skills, to dissociate the neurophysiological mechanisms underlying the self-generated versus externally imposed actions, and to investigate an error detection mechanism that monitored whether the final sensory feedback was coherent with the expected sensory consequences of actions [34]. In Experiment 1, participants performed a flanker task, which required a quick response to left or right arrows at the center of a stimulus array with the presence of surrounding flankers, either compatible or conflicting with the central arrow direction. Notably, some trials were intentionally incongruent with participants’ intended movements. In Experiment 2, participants observed the virtual avatar while it performed correct and incorrect movements during the same task. The results showed that the sense of agency was lower in incongruent conditions than in congruent conditions. Regarding the neurophysiological mechanisms behind actions generated in the first person as compared to those imposed from the outside (third person), a greater amplitude of frontocentral negativity error with a latency of 100 ms (Ne/ERN) was recorded during “first-person errors”, whereas a later parietal negative component (N400), related to “third-person errors”, was detected, its amplitude negatively correlated with a sense of ownership.

The work of Metcalfe and colleagues aimed at using “turbulence” (spatial asynchrony between the position of the mouse and the cursor on the display) in a skill-learning task [30]. They performed two different experiments to investigate the impact of variations in proximal actions (finger pressing the button) and distal outcomes (a light turning on) on people’s judgments of agency and of performance (using a questionnaire) in which participants had to identify (by pressing in Experiment 1 and by exploding in Experiment 2) specific targets and avoid others. Participants reported higher scores for the judgment of agency and performance in the absence of turbulence conditions in both experiments; moreover, turbulence decreased the judgment of performance and the performance success rate (hit and explode).

##### Temporal Mismatch

Rognini and colleagues used the cross-modal congruency effect (CCE, a paradigm that quantifies perceptual integration [40]) in a task requiring circular movements using a bimanual haptic interface in three experiments [36]. The virtual reality task comprised visual distractors that could appear in different locations and asynchronously to the haptic feedback. In Experiment 1, participants were asked not to move their hands; in Experiment 2, participants performed slow clockwise circular movements, and they were trained for static and movement conditions; in Experiment 3, participants were trained for synchronous and asynchronous conditions. In Experiment 1, CCEs (incongruent errors minus congruent errors) in the static condition as compared to the movement condition were significantly larger in same-side conditions than in different-side conditions, whereas the reaction times showed a main effect of congruency and a significant interaction between side and congruence. In Experiment 2, CCEs in the static conditions were significantly larger than in the movement conditions. In Experiment 3, the amplitude of CCEs in the same-side condition was significantly larger in the synchronous than in the asynchronous condition, and this result could lead back to a reduced sense of agency perceived in the asynchronous conditions.

Temporal delay and physical mismatch (tremor) were introduced by Nataraj and colleagues to study the association between agency (measured using an intentional binding and self-report questionnaire) and performance in a reaching task [32]. Significant differences in both agency and performance across different types of control modes (speed modifications, addition of mild noise, and automation) were found, with higher values of agency and performance during the match between observed movement and real movement. Moreover, positive correlations between agency and reaching performance indices were found.

A temporal mismatch of auditory feedback was used by Tidoni and colleagues who aimed to demonstrate the importance of auditory feedback in the telemanipulation of a humanoid robot and the consequences of its distortion [37]. The subjects controlled the robot in four conditions: synchrony (footstep sound was synchronous with the robot’s footsteps), asynchrony (footstep sound was asynchronous), the presence of a mirror (subjects saw the reflection of the robot in the mirror), and the absence of a mirror (subjects could not see the reflection of the robot). There was no difference either in sense of agency between different conditions or in learning. The authors reported an effect of auditory feedback, with a faster time in the synchronous condition relative to the asynchronous condition, but no effect of visual feedback.

Weibel and colleagues investigated whether the rise in the feeling of control could be related to the existence of small and undetected distortions in haptic feedback using a task comprising small temporal delays [38]. Participants performed a virtual pointing task using a robot with haptic feedback, sometimes postponed by a small temporal delay (15 or 65 ms), making the haptic distortion explicit and implicit (Experiment 1) or implicit (Experiment 2). The results showed that, after the distortions, the hand trajectories were effectively adapted, but the feeling of agency significantly diminished in both experiments (with implicit and explicit distortion). Experiment 1 did not show a significant effect of the distortion on the amplitude of deceleration, whereas Experiment 2 did (deceleration was lower in the trial following distortion). There was no significant effect of the distortion on the amplitude of deceleration, but a significant interaction was found between distortion delay and rank (degree of distortion) when compared with the conditions without distortion. A higher feeling of agency was perceived when no distortion occurred, as compared to other conditions.

##### Physical Mismatch

Kumar and Srinivasan aimed to investigate how control at multiple levels influences agency. The authors employed intentional binding, in which participants must estimate the time interval between pressure on a trigger and the appearance of a circle on the screen, a measure of agency [29]. The task was made more difficult by adding noise to the joystick controller, resulting in different degrees of control. The results showed that the sense of agency increased with the amount of joystick control for incorrect hits, whereas no differences were observed for correct hits.

Ozen and colleagues investigated motor learning using haptic mismatch with the main aim of testing the use of model predictive controllers (MPCs), a versatile control methodology wherein a system model is leveraged to predict the future behavior of the controlled system [33]. The control inputs are determined by optimizing a cost function defined by the user, considering that the predicted information obtained from the system model could be employed in motor learning settings [41]. Participants had to swing a virtual pendulum to hit targets while holding the end-effectors of a robot. Two MPCs with two different points of force application were used: application on the effectors, (i.e., controllers, “eeMPC” condition) or directly on the pendulum (“ballMPC” condition). A subjective measure of intrinsic motivation was administered after the task. The results showed a significant effect of the type of training on agency; training with ballMPC reported lower levels of agency than training without MPC. Despite the greater force applied in the eeMPC condition, there was no significant effect of the training modality on the questionnaire assessing intrinsic motivation.

Aoyagi and colleagues altered subjective motor control via spatial, temporal, and physical mismatch during a task requiring the subject to obtain a match between virtual objects and their hand (measured by motion-monitoring sensors) [26]. In both experiments, agency was significantly impacted by visual modification and weight. In Experiment 1, motion error was greater in the offset condition, where the visual stimulus was positioned at the midpoint between the real hand position and the nearest point on the trajectory. The visual prediction error was smaller compared to the aligned condition, leading to higher agency perception rated by participants. Despite the larger motion error in the offset condition, the visual prediction error was smaller, contributing to higher agency ratings. In Experiment 2, the influence of weight on the sense of agency was also significant. The comparison between the “weighted and aligned” and “weighted and offset” conditions revealed a significant impact of visual stimulus modification. The rate of visual prediction errors was considerably lower in the “weighted and offset” condition as compared to the “weighted and aligned” condition.

#### 3.2.2. Studies Based on Experimental Feedback

Five studies employed different types of feedback to analyze their impact on the perceived sense of agency and skills learning (Table 3, Figure 2B, Appendix A) [25,27,29,31,39]. Among these studies, two main experimental categories could be identified:The administration of different types of sensory feedback during action execution, involving unimodal, bimodal, or multimodal derogation;The administration of subjective performance feedback, specifically designed to provide information about the individuals’ task abilities.

##### Sensory Feedback

Ozen and colleagues used model predictive controllers (MPCs) to investigate how different assistance strategies (interventions aimed at creating different degrees of control) impacted the sense of agency and the learning of a complex dynamic task using an interface providing haptic feedback [39]. To this aim, participants were tasked with swinging a virtual pendulum to strike incoming targets with the pendulum ball. They were randomly divided into four groups: control, end-effector MPC (applying assisting forces directly to the end-effector, thereby directly influencing pendulum dynamics and being less susceptible to human-robot interaction forces), ball MPC (applied forces directly to the virtual pendulum ball), and haptic guidance (using a conventional haptic guidance controller with fixed trajectories). The authors hypothesized that the indirect application of the assisting forces could reduce the participants’ sense of agency because of a lower perception of the pendulum movements; this could lead to motor learning difficulties. The results of the end-effector MPC group showed an augmented interaction force between the participant and robot end-effector (which monitors the controller’s endpoint during training), increased assisting force (applied by the different controllers at the robot end-effector), and improvement in the overall performance, thus better controlling the pendulum. In addition, a positive correlation between the amount of learning and the perceived sense of agency was observed in the end-effector MPC group.

A similar visuomotor task was used by Haar and colleagues, which employed a billiards paradigm allowing for control of the visual feedback for the real-world task while maintaining the sense of embodiment in virtual reality and the real world (matched and presented simultaneously), during which participants received visual and tactile feedback [27]. The results showed the following differences in a virtual environment compared to real-world tasks: a general subjective improvement in success rates at the cost of slower learning; a gradual reduction in mean directional absolute error, although slower and less pronounced; a decrease in the velocity profile error and the intertrial variability over learning.

Regarding the investigation of behavioral and psychophysiological measures of the sense of agency and skills learning during the administration of visual and haptic feedback, the study of Adamovich and colleagues aimed at delineating the brain–behavior interactions in virtual reality using functional magnetic resonance imaging during a finger movement task [25]. Participants wore a glove to control a virtual hand model and were exposed to four different conditions, where they either observed with the intent to imitate or execute finger sequences performed by a virtual hand avatar. The results showed that both observing to imitate and receiving real-time feedback from the virtual avatar led to activation in a distributed frontoparietal network. Moreover, a time-dependent increase in activation was also detected in the left insular cortex during observation with the intent to imitate actions performed by the virtual avatar. Imitation with virtual avatar feedback, when compared to a control condition, elicited activations in brain regions associated with the sense of agency, including the angular gyrus, precuneus, and extrastriate body area.

##### Performance Feedback

Nataraj and colleagues delivered feedback based on subjective performance (positive or negative feedback according to the assignment group) to investigate its impact on performance (reaching a blue target on the panel), sense of agency, and locus of control towards actions [31]. The results showed that the performance and the sense of agency increased in the group receiving positive feedback when compared to the group receiving negative feedback, but no changes were detected regarding the locus of control. Moreover, during positive feedback training, there was a progressive increase in the sense of agency, accompanied by a decrease in performance.

Similarly, Kumar and Srinivasan’s Experiment 2 investigated the impact of two performance feedback (introduced to manipulate subjective control at the goal level) on agency (measured using an intentional binding task) [29]. Participants were supposed to estimate the time interval between pressing a trigger and a circle flashing on the screen during the administration of feedback about their performance, either during the action (trigger pressing) or after observing the outcome. When the feedback coincided with actions, a main effect indicating a lower estimated interval of 400 ms as compared to 700 ms was detected alongside a significant main effect of target completion, with larger estimations when the target was completed than when the target was missed. However, when the feedback was presented after observing the outcome, an overlapping main effect of the sense of agency was observed, but no significant effect of target completion was detected. These findings indicate that subjects’ estimates of duration were influenced by their knowledge of the task’s goal.

## 4. Discussion

The sense of embodiment allows for the integration of external tools in the subjective body’s self-representation. The motor component of embodiment, the sense of agency, permits the experience of global motor control, including action, control, intention, motor selection, and the conscious experience of will [42]. This ability is pivotal in the remote operation field, as it promotes skills-learning processes that increase confidence towards the tools to be manipulated, thus minimizing inconsistencies in size, movement, and degrees of freedom of the teleoperated device [3].

Herein we have collected and analyzed the results of the current scientific literature on the relation between the sense of agency and skills-learning ability in teleoperation environments.

### 4.1. Mismatch Diminishes the Sense of Perceived Agency and Task Performance during Action Execution

Spatial mismatch highlights the importance of spatial synchrony in enhancing agency perception, with a dissociation observed between embodiment components [34,35]. The sense of agency appears to be more influenced by the synchrony of the motor action (also supported by the higher amplitude of the N400, usually associated with error detection), while ownership depends more on the appearance of the tools [34,35]. This confirms that agency is heavily influenced by congruence between intentions and actions, in contrast to ownership. This dissociation is also supported by Metcalfe and colleagues, who found that the mismatch affecting the outcome of the actions altered the sense of agency [30]. When the outcome was not the intended goal, the subjects felt a decrease in perceived performance and a sense of control. When individuals have higher control over movements, the outcome of actions follows the individuals’ intentions independently of their appearance [30]. This would therefore explain why agency is more influenced by the congruence between the action and the result and less by appearance and location, differently from ownership. A different contribution came from the study by Kannape and colleagues, which demonstrated that inconsistency between the subject’s body and a virtual avatar negatively affects performance but not awareness of movement; it can be assumed that, despite the lack of awareness of some body parts, subjects can maintain awareness of the position and locomotion of the entire body, preserving the sense of self-hood [28]. On the contrary, when incongruity between executed and seen movement overcomes a certain threshold of mismatch, the subject consciously readjusts the parameters during task execution [28]. Aoyagi and colleagues showed that introducing modified visual feedback can reverse the decrease in agency perception caused by temporal perturbations, thus highlighting that the introduction of visual feedback that matches the motor intention may enhance both the individual’s sense of agency and their effectiveness in planning and selecting the subsequent motor steps [26].

The sense of agency relies more on spatial congruence between executed and observed movement, while ownership is influenced by physical appearance [43], and the hypothesis that agency arises from synchrony between intended and executed actions is corroborated by the increased sense of agency in congruence conditions [34]. These findings suggest that a match between programmed and executed actions enhances agency perception and motor control [26,28,29,30,32,33,34,35,36,37,38].

Other studies have used temporal mismatch, in which a temporal delay is experimentally introduced between executed and seen movements. The temporal delay is typically introduced to create an incongruence between visual, auditory, and/or haptic information. Most studies found a negative impact on the sense of agency and task performance ascribable to temporal delays [32,36,38]. However, Aoyagi and colleagues and Tidoni and colleagues reported no effect on agency perception [26,37]. Aoyagi and colleagues attributed this to the introduction of delayed visual feedback, which reduces prediction errors and enhances agency perception [26]. Tidoni and colleagues administered auditory asynchrony applied to avatar footsteps; while the visual feedback introduced by the authors maintained an adequate sense of agency, the auditory asynchrony without visual feedback negatively impacted skills learning, thus worsening motor performance [37]. This suggests that despite temporal delays, agency perception can be preserved if the match between the planned action and the consequence is maintained. However, repeated temporal asynchrony experiences may hinder motor program preparation, preventing the emergence of agency perception and negatively affecting skills learning [38,44].

Studies on physical mismatch found a decrease in the sense of agency and skills learning after the mismatch of the virtual object (pendulum instead of robotic arm effectors), or after the physical noise applied to the movement of the virtual hands [32,33]. The decrease in agency perception may stem from a feeling of “false movement” when forces are applied to objects, preventing the subjective perception of reproducing those forces [33]. Differently, Aoyagi and colleagues (Experiment 2) reported no difference in motor precision but noted a decreased sense of agency after applying weight to participants’ wrists [26], thus demonstrating that minor adjustments to compensate for physical mismatch mitigated its impact, augmenting participants’ sense of agency [26]. Kumar and Srinivasan (Experiment 1) reported that the perceived time interval between action and outcome was longer in low-control conditions, indicating different impacts on agency related to control level: at the perceptual–motor level, the sense of agency is supposed to be influenced during incorrect trials, a condition that in turn implies a low perceived control (e.g., joystick control) at the higher goal level, as subjects failed to achieve the goal during the task execution [29]. These studies suggest that the degree of mismatch, rather than its mere presence, influences agency perception, potentially interfering with motor programming [45], thus highlighting that agency arises not only from comparing programmed and executed actions but also from the accurate execution of motor programs, which involves the continuous updating of predictions [45].

In summary, the pool of articles included in this review highlights how different types of mismatch reduce the individual’s perception of control towards actions, thus leading to poorer performance during tasks, probably owing to the mediating role of the sense of agency. Also, the impact of a higher degree of mismatch could impede an optimal method of skills learning, which could be slower, less effective, or even prevented. Nevertheless, it is essential to recognize that the central nervous system seems to preserve the sense of agency (also by the integration of different stimuli) when mismatch remains below a certain threshold; this is necessary to ensure congruence between planned and executed actions, together with correct motor programming, thus allowing the maintenance of the sense of agency.

### 4.2. The Sense of Agency Is Increased by Multisensory Integration and Augments Skills Learning

Several studies have emphasized the importance of multisensory integration in teleoperation environments for enhancing the sense of agency and performance, thus supporting its role in producing a virtual experience as realistically (i.e., similar to a physical environment) as possible [25,27,29,31,39]. In this line, various types of feedback were administered to identify changes in the sense of agency and skills performance.

#### 4.2.1. The Modulation of the Sense of Agency Varies According to the Number and the Timing of Feedback

Adamovich and colleagues [25] studied the impact of visual feedback on neurophysiological correlates of agency during skills learning, finding that the imitation of virtual avatar feedback (relative to the control condition) was associated with the activation of regions involved in the sense of agency, such as the angular gyrus, precuneus, and extrastriate body area, as well as the right cerebellum. The interpretation of this result is based on the model that involves the integration between visual and proprioceptive feedback with motor commands. This visuomotor integration confirmed the previous results of Padrao and colleagues, who found a larger distribution of the ERP component N400 (indicating an error detection process) in parietal sites; this led the authors to suggest that these areas may be involved in error monitoring during voluntary action [34].

The effect of bimodal feedback on agency has produced heterogeneous results [27,39]. A positive correlation between agency and skill acquisition was observed in less skilled participants [39], whereas the use of feedback in virtual reality led to slower but more effective learning compared to real-world learning [27]. Despite the use of multimodal feedback to recreate a physical reality, multisensory integration in teleoperation settings may be limited, leading to reliance on specific sensory modalities [46]. While multisensory feedback can augment embodiment and performance in less skilled individuals, its artificial nature may reduce agency perception, especially in skilled individuals. Indeed, skilled participants may perceive assisting forces as disturbances, reducing their sense of agency [47,48].

Regarding performance feedback, positive performance feedback supports visuomotor performance and agency perception [29,31], potentially enhancing motivation and confidence. Agency appears to be influenced by hierarchical processes, with various forms of feedback assisting individuals at different skill-learning stages. Crucially, the level of confidence in actions is intimately related to a hierarchical predictive control system that forms the bedrock of perception, action, and consciousness [49,50,51,52]. In conclusion, feedback can increase agency and performance, but its effectiveness may depend on two crucial factors: participants’ skill level and the timing of administration during action execution.

#### 4.2.2. Is There a Hierarchy among Sensory Modalities in Promoting a Sense of Agency and Skills Learning? No, but Multisensory Integration Is Essential

Among the studies included, there is no agreement on the predominance of one type of sensory stimuli in supporting the sense of agency, but instead, it highlights the pivotal role of multisensory integration in enhancing performance within a teleoperation environment. It has been pointed out that discrepancies interfering with the sense of ownership occur when visual feedback is provided asynchronously while tactile feedback and proprioceptive output remain synchronous, whereas the sense of agency is sensitive to the temporal asynchrony between the proprioceptive component of action execution and visual feedback. Such temporal misalignments may undermine participants’ perceived control over displayed movements and attenuate their sense of embodiment [53].

In this line, one may ask whether one sensory modality has a greater impact on agency than the others. It has been argued that each sensory modality may exert non-hierarchical influence over embodiment and agency in teleoperation environments, with multisensory integration involving the cohesive process of incorporating input from all senses [54]. These signals collectively form a mental representation of distant stimuli, perceived as a unified whole (the percept) [54]. When this integration process is disrupted, usually due to diminished somatosensory feedback or sensory–motor discrepancies, perceptual incoherence, resulting from conflicting or imbalanced sensory inputs, arises; this prevents the integration of information into a cohesive percept, resulting in fragmented experiences [54].

To prevent this phenomenon, it is obligatory to ensure perceptual coherence in teleoperation environments; this will prevent negative effects on the perception of self and the sense of agency during action execution. Following Cesari and colleagues, the use of immersive and augmented reality holds promise for augmenting multisensory integration and perceptual coherence, potentially enhancing embodiment, flow, and telepresence [55]. The incorporation of diverse forms of multisensory feedback may offer a more natural experience, facilitating more effective learning within virtual and teleoperation environments.

## 5. Limitations

Although the present work employs a rigorous methodology, it has some limitations that need to be addressed. Firstly, the small number of studies included (N = 15) makes it difficult to draw robust conclusions. Additionally, small sample sizes are prevalent within each study, particularly in between-study designs. Other important issues concern the heterogeneity of the included studies, which prevents a meta-analytic comparison. In fact, experimental paradigms and the type of agency measure (implicit; explicit; mixed) were differently represented among the included studies. Despite the inherent heterogeneity, our systematic approach of grouping (mismatch vs. feedback) and subgrouping (mismatch: spatial, temporal, and physical; experimental feedback: sensory and performance) based on experimental paradigms effectively mitigated the bias associated with heterogeneity, fostering a nuanced and interpretable cross-study analysis.

## 6. Conclusions

The sense of agency enables the experience of motor control, a crucial ability in the field of teleoperation. It increases confidence in manipulating the tools thanks to its involvement in dexterity performance. It can be hypothesized that the greater the sense of agency, the greater the ability to learn skills.

This systematic review is the first to highlight the relationship between the sense of agency and the ability to learn skills in teleoperation environments, thus enriching this line of research by integrating qualitative information from research fields. The results provide an in-depth understanding of the sense of agency, taking into account the impact of different types of experimental interventions (mismatch or experimental feedback) on the sense of agency and learning. The degree to which motor information depends on mismatch is proportional to the negative impact on the sense of agency and learning due to greater interference with motor programming. The sense of agency and skills learning benefit from sensory feedback, whether it is unimodal, bimodal, or multimodal. However, the effect of feedback on agency and learning is closely dependent on the participant’s skill level and experience. The artificial nature of feedback, as compared to feedback in a natural environment, could decrease the sense of agency and learning. In addition, feedback timing is based on control at either the goal level (achievement of task goal) or at the perceptual–motor level (tool control), resulting in increased confidence and motivation to act. Importantly, the beneficial effect of feedback on agency and skills learning is not attributed to a preferential sensory modality. This is because multisensory integration appears to be the most relevant factor.

This work could shed light on all the teleoperation settings where the sense of agency is crucial for skills learning, such as robotic-assisted surgery, where the degree of confidence towards robotic arms could facilitate surgical outcomes.

Although the experimental tasks in the included studies may lack ecological validity, it is worth noting that the introduction of several experimental perturbations (such as mismatches) or augmentations (such as sensory and performance feedback) might be helpful for a better understanding of teleoperation settings. For robotic surgery, managing temporal delay in communication between the surgeon and the robotic arms is critical, especially for long-distance operations [56,57,58,59]. The included studies explore the effect of temporal asynchrony on agency and visuomotor skills, offering a new perspective on managing this issue. This work might also address the introduction of sensory feedback, particularly haptic feedback, during robotic-assisted operations, which is extensively studied and widely debated. The use of haptic feedback has proven to be crucial for various surgical competencies, including the refinement of force control, advancement of patient safety, and reduction of surgeon fatigue during operations [60]. This holds particularly true for novice surgeons, who argue that haptic feedback could enhance learning and reduce workload during robotic training, especially in the early stages [60,61]. In this line, our contribution may help to solve this conundrum by examining the role of feedback on the sense of agency, rather than limiting the focus to mere performance.

## Figures and Tables

**Figure 1 brainsci-14-00350-f001:**
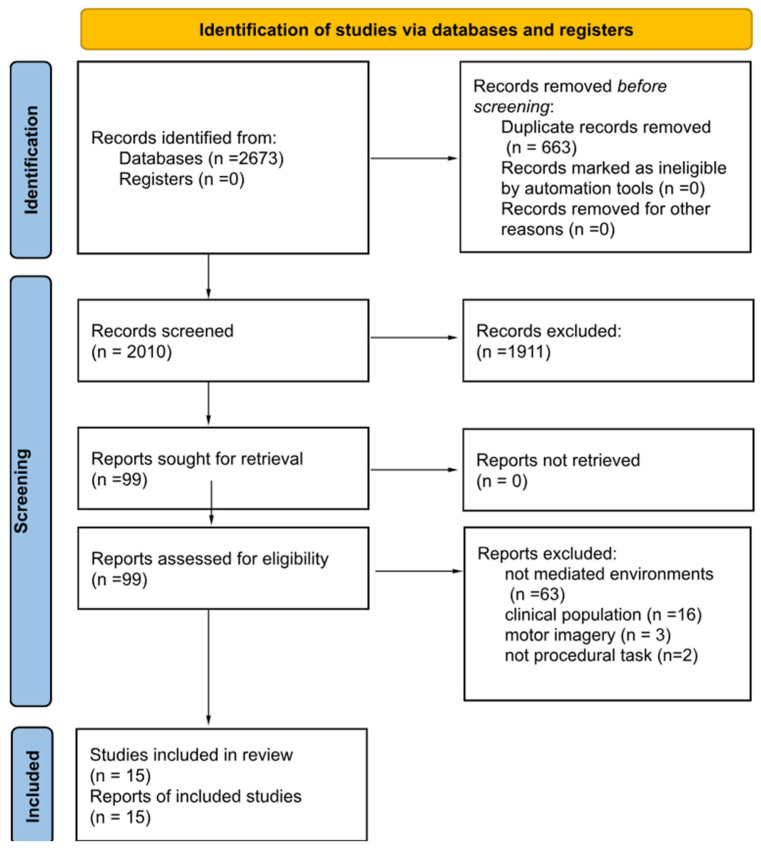
Flow chart of included studies.

**Figure 2 brainsci-14-00350-f002:**
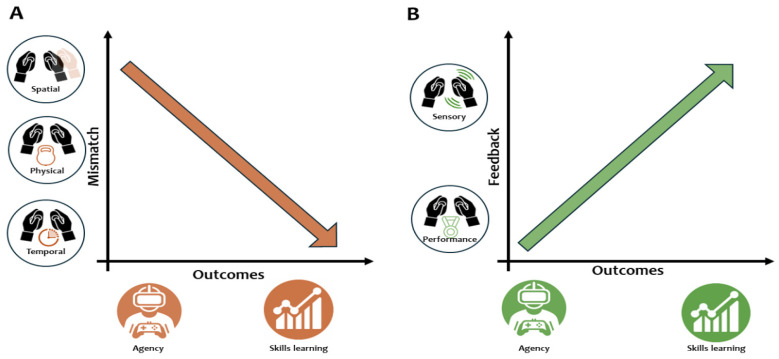
Synthesis of the main findings. (**A**): Introducing mismatches (spatial; physical; temporal) decreases the sense of agency and performance. (**B**): The introduction of feedback (sensory; performance) increases the sense of agency and performance.

**Table 1 brainsci-14-00350-t001:** Population, Intervention, Comparison, Outcomes, and Study Design (PICOS).

Parameter	Inclusion Criteria	Exclusion Criteria
Participants	Healthy subjects aged ≥ 18 years	Subjects belonging to the clinical population (amputees, patients with vestibular disorders, patients with prostheses, etc.)
Interventions	Manipulation of virtual reality settings for the assessment of skills-learning tasks and sense of agency	Absence of virtual reality settings and manipulation; absence of behavioral or psychometric indices; skills-learning tasks performed using motor imagery
Comparisons	Within-subject or between-subjects changes in skills-learning performance and sense of agency (explicit or implicit) due to manipulation of virtual reality settings (absence/presence of multimodal feedback, absence/presence of perturbations, before/after training, virtual reality/real world, and humanoid/non-humanoid robots)	Comparison of skills-learning performance and sense of agency (explicit or implicit) with the absence of manipulation of virtual reality settings
Outcomes	Behavioral, psychometric, and/or mixed (behavioral and physiological) indices of skills learning and agency	Behavioral, psychometric, and physiological parameters of no interest, nor methodological issues related to the collection of such parameters
Study design	Within subjects, between subjects, crossover, and pre/post	Case reports, methodological issues, and lack of replicability

**Table 2 brainsci-14-00350-t002:** Main findings as a function of mismatches.

Reference	Type of Mismatch	Findings
Aoyagi et al., 2021 [26]	Spatial, Temporal, Physical	The mismatch between virtual object movements and hand movements reduced control. When virtual objects were slightly offset, the feeling of control was augmented despite performance errors. Weights applied to participants’ hands influenced the sense of control.
Kannape et al., 2010 [28]	Spatial	The mismatch in perceived and actual movement influenced walking direction. Enhanced performance occurred when viewing the virtual body from specific angles.
Kumar and Srinivasan, 2017 [29]	Physical	The sense of agency was higher as participants felt greater control over the joystick. This effect was notably pronounced for incorrect hits.
Metcalfe et al., 2013 [30]	Spatial	The absence of turbulence resulted in higher perceived agency and skills learning.
Nataraj et al., 2020b [32]	Temporal/Physical	Higher values of agency and performance were observed during synchrony between participants’ movement and virtual movement, with a positive association between agency and reaching performance indices.
Ozen et al., 2019 [33]	Physical	Differences in agency perception based on the type of training modality employed (control mode). Reduced agency was reported when utilizing the pendulum control modality.
Padrao et al., 2015 [34]	Spatial	A mismatch between participants’ movements and visual feedback reduced the feeling of control. Self-generated errors led to frontocentral negativity with a 100 ms latency, while externally imposed errors led to the N400.
Ratcliffe and Newport, 2017 [35]	Spatial	Agency was higher during the synchrony of the movement, while ownership was also influenced by appearance, location.
Rognini et al., 2013 [36]	Temporal	Larger cross-modal congruency effects were associated with a reduced sense of agency in the asynchronous condition.
Tidoni et al., 2014 [37]	Temporal	Synchronous auditory feedback (footsteps) resulted in faster skills learning.
Weibel et al., 2015 [38]	Temporal	Perceived agency decreased when there were minor time delays.

**Table 3 brainsci-14-00350-t003:** Main findings as a function of experimental feedback.

Reference	Type of Feedback	Findings
Adamovich et al., 2009 [25]	Sensory	Real-time feedback activated a distributed frontoparietal network, with imitation of action leading to activations in regions sustaining the sense of agency.
Haar et al., 2021 [27]	Sensory	Haptic and visual feedback delivered during the virtual reality billiard task led to subjective improvement in success rates but slower learning compared to real-world tasks.
Kumar and Srinivasan, 2017 [29]	Performance	Performance feedback led to shorter estimated intervals (elapsed time between pressing the trigger and the circle).Performance feedback presented after observing the outcome showed an overlapping main effect of agency.
Nataraj et al., 2020a [31]	Performance	Positive feedback led to better performance and a greater sense of agency.
Ozen et al., 2021 [39]	Sensory	Applying assisting force feedback on end-effectors showed better performance and a positive correlation between learning and perceived agency.

## Data Availability

No new data were created. Data sharing is not applicable to this article.

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
