# Peer review of "Sense of Agency and Skills Learning in Virtual-Mediated Environment: A Systematic Review"

_brainsci, 2024, doi:10.3390/brainsci14040350_

Round 1

Reviewer 1 Report

Comments and Suggestions for Authors

Sense of agency and skills learning in virtual-mediated environment:      a systematic review

Submission to MDPI Brain Sciences

(1) General comments:

The structure and content are alright. I wonder if ‘multisensory stimulation’ is a relevant keyword. Minor edits in language are recommended. I find the content is too wordy. I suggest the authors provide a summary schematic diagram related to the Results section (type of mismatch and type of feedback).

(2) Introduction:

The authors described in greater detail the concepts of agency, different types of agency, and so on. Please provide 1-2 examples of applications or use cases of the environment, which is the focus of this article. The authors may change the location of this in the ‘Introduction’ instead of in the ‘Discussion’. This is because the beginning of the manuscript shall explain what and why the environment is of interest. This is the gap to fill or the selling point of this work.

(3) Methods:

The article uses PRISMA guideline.

(4) Results:

I recommend Table 1 to be in the landscape mode for readability. The authors have written a good synthesis of the literature. I appreciate the types of sensory/performance feedback discussed.

Can the authors summarize the putative neural substrates involved in such an environment? I strongly recommend this, as this is Brain Sciences journal.

Comments on the Quality of English Language

Proof-reading required.

Reviewer 2 Report

Comments and Suggestions for Authors

Dear Authors,

Sense of agency and skills learning in virtual-mediated environment: a systematic review

in the introduction :It would have been better if the authors mentioned the biological and physiological basis for the research of this review It would have been better if the authors mentioned the biological and physiological basis for the research of this review

The authors did not mention the following points in detail

Criteria for considering studies for this review

Types of studies

Types of participants

Types of interventions

Types of outcome measures

Search methods for identification of studies

Effects of interventions

The statistical study is clear and the tables are appropriate and explain the results.

. The discussion section is Suitable and explain many of the results

-References and citations are appropriate. up to date and their number is appropriate

many thanks
